# Ferromagnetism above 1000 K in a highly cation-ordered double-perovskite insulator $Sr_3OsO_6$

Yuki K. Wakabayashi[1], Yoshiharu Krockenberger[1], Naoto Tsujimoto[2], Tommy Boykin[1], Shinji Tsuneyuki[2], Yoshitaka Taniyasu[1] & Hideki Yamamoto [1]

Magnetic insulators have wide-ranging applications, including microwave devices, permanent magnets and future spintronic devices. However, the record Curie temperature ($T_C$), which determines the temperature range in which any ferri/ferromagnetic system remains stable, has stood still for over eight decades. Here we report that a highly B-site ordered cubic double-perovskite insulator, $Sr_3OsO_6$, has the highest $T_C$ (of ~1060 K) among all insulators and oxides; also, this is the highest magnetic ordering temperature in any compound without $3d$ transition elements. The cubic B-site ordering is confirmed by atomic-resolution scanning transmission electron microscopy. The electronic structure calculations elucidate a ferromagnetic insulating state with $J_{eff} = 3/2$ driven by the large spin-orbit coupling of $Os^{6+}$ $5d^2$ orbitals. Moreover, the $Sr_3OsO_6$ films are epitaxially grown on $SrTiO_3$ substrates, suggesting that they are compatible with device fabrication processes and thus promising for spintronic applications.

[1] NTT Basic Research Laboratories, NTT Corporation, 3-1 Morinosato Wakamiya, Atsugi, Kanagawa 243-0198, Japan. [2] Department of Physics, The University of Tokyo, 7-3-1 Hongo, Bunkyo-ku, Tokyo 113-0033, Japan. Correspondence and requests for materials should be addressed to Y.K.W. (email: yuuki.wakabayashi.we@hco.ntt.co.jp)

**M**agnetic insulators have been intensively studied for over 100 years, and they, in particular ferrites, are considered to be the cradle of magnetic exchange interactions in solids. Their wide range of applications include microwave devices[1] and permanent magnets[2]. They are also suitable for spintronic devices owing to their high resistivity[3], low magnetic damping[4], and spin-dependent tunneling probabilities[5]. The Curie temperature ($T_C$) is the crucial factor determining the temperature range in which any ferri/ferromagnetic system remains stable. However, the record $T_C$ has stood still for over eight decades in insulators and oxides (943 K for spinel ferrite $LiFe_5O_8$)[6].

The B-site ordered double-perovskite $A_2BB'O_6$ family includes lots of fascinating magnetic materials such as half-metals[7–9], multiferroic materials[10], antiferromagnetic (AFM) materials[11], and magnetic insulators[12–14]. The A site is usually occupied by an alkaline-earth or rare-earth element, and B and B' are transition metal elements. Explorations of magnetism have mainly focused on varying the combination of transition metal elements at B and B' sites, and it has been believed that having them occupied by two different transition metal elements is a prerequisite for a magnetic order at high temperatures[8]. Some $4d$ or $5d$ element-containing double-perovskites, e.g., $Sr_2FeMoO_6$[7] ($T_C = 415$ K), $Sr_2CrReO_6$[9] ($T_C = 634$ K), and $Sr_2CrOsO_6$[13] ($T_C = 725$ K), reach a point of ferromagnetic (FM) instability at high temperatures, although the majority of double-perovskites show an AFM order or weak spin-glass behavior[15].

In this work, we show that a highly B-site ordered double-perovskite insulator, $Sr_3OsO_6$, surpasses the long-standing $T_C$ record by more than 100 K. In contrast to other $4d$ or $5d$ double-perovskites that follow the above-mentioned criteria, we discovered FM ordering above 1000 K in $Sr_3OsO_6$, in which only one $5d$ transition metal element occupies the B sites. Remarkably, the $T_C$ of $Sr_3OsO_6$ (~1060 K) is about ten times higher than the previous highest magnetic transition temperature in double-perovskites including only one transition element ($Sr_2MgOsO_6$, AFM, $T_N = 110$ K)[16]. We revealed this B-site ordering using atomic-resolution scanning transmission electron microscopy. The density functional theory (DFT) calculations suggest that the large spin–orbit coupling (SOC) of $Os^{6+}$ $5d^2$ orbitals drives the system toward a $J_{eff} = 3/2$ FM insulating state[17–19]. Moreover, the $Sr_3OsO_6$ is the epitaxially grown osmate, which connotes that it is compatible with device fabrication processes and thus promising for spintronic applications.

## Results

**Epitaxial growth and crystallographic analyses.** High-quality single-crystalline B-site-ordered double-perovskite $Sr_3OsO_6$ films (300-nm thick) were epitaxially grown on (001) $SrTiO_3$ substrates with an abrupt substrate/film interface in a custom-designed molecular beam epitaxy (MBE) setup capable of precisely controlling elemental fluxes even for elements with high melting points, e.g., Os (3033 °C) (Methods). Maintaining a precise flux rate for each constituent cation (Os and Sr) with a simultaneous supply of $O_3$ is essential for avoiding deterioration of the magnetic properties as a deviation of only 2% from the optimal Os/Sr ratio is fatal (Methods).

High-resolution scanning transmission electron microscopy (STEM) and transmission electron diffraction (TED), combined with high-resolution reciprocal space mapping (HRRSM) and reflection high-energy electron diffraction (RHEED), ascertained a cubic double-perovskite structure[8,12,20] (Methods). As schematically shown in Fig. 1a, d (viewed along [100] and [110] directions), Sr- or Os-occupied, fully Sr-occupied, fully Os-occupied, and fully oxygen-occupied columns exist. The STEM images overtly demonstrate that these columns are arranged in a spatially ordered sequence. Since the intensity in the high-angle annular dark-field (HAADF)-STEM image is proportional to $\sim Z^n$ ($n \sim 1.7$–2.0, and Z is the atomic number), in Fig. 1b, the white spheres and gray ones are assigned to Sr- ($Z = 38$) or Os- ($Z = 76$) occupied and fully Sr-occupied columns, respectively. In Fig. 1e, the white spheres and gray ones are assigned to fully Os-occupied and fully Sr-occupied columns, respectively. In contrast to HAADF-STEM, Oxygen is emphasized in annular bright-field (ABF)-STEM images. Accordingly, Oxygen ions occupying the

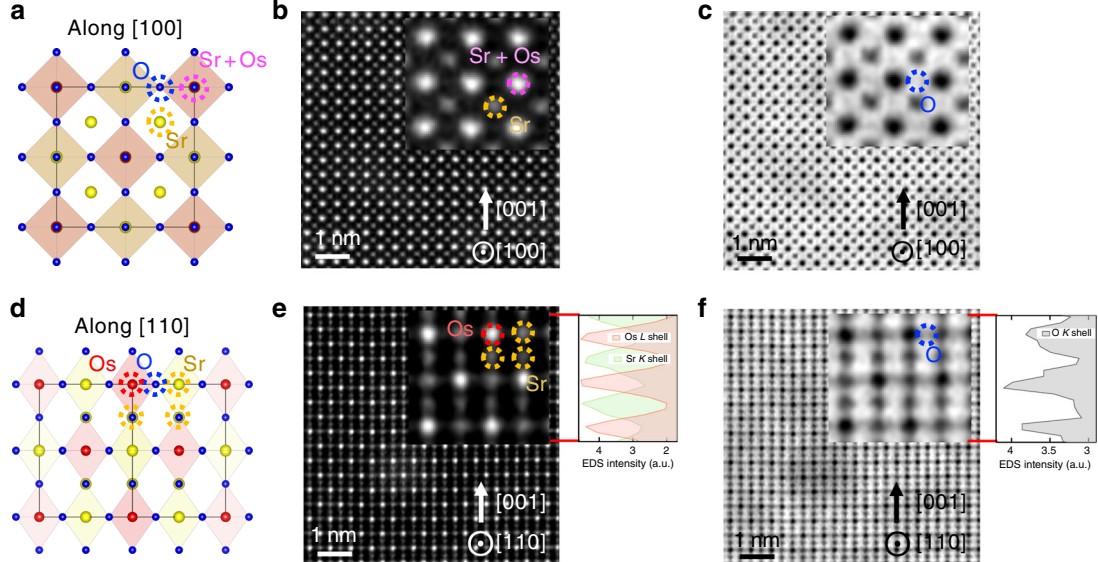

**Fig. 1** Atomic-resolution STEM images of a $Sr_3OsO_6$ film. **a** Schematic diagram of the $Sr_3OsO_6$ viewed along the [100] direction. **b**, **c** HAADF-STEM (**b**) and ABF-STEM (**c**) images near the center of the $Sr_3OsO_6$ layer along the [100] direction. **d** Schematic diagram of the $Sr_3OsO_6$ viewed along the [110] direction. **e**, **f** HAADF-STEM (**e**) and ABF-STEM (**f**) images near the center of the $Sr_3OsO_6$ layer along the [110] direction. The scale bars are 1 nm. The insets in **b**, **c**, **e**, and **f** show enlarged views together with EDS-STEM intensity profiles along the [001] direction. In all figures, purple, yellow, red, and blue dotted circles indicate Sr- or Os-occupied, fully Sr-occupied, fully Os-occupied, and fully oxygen-occupied columns, respectively

expected positions are clearly visible (labeled O (insets of Fig. 1c, f)). The energy dispersive X-ray spectroscopy (EDS)-STEM intensity profiles along the [001] direction shown in Fig. 1e and f complementarily confirm the above elemental assignments. The peak positions in the EDS profile of the Os $L$ shell, Sr $K$ shell, and oxygen $K$ shell agree well with the Os, Sr, and oxygen positions, respectively, determined by STEM. The STEM observation revealed the rock-salt type order of $Os^{6+}$, whereas the hexavalent state of Os is confirmed by X-ray photoemission spectroscopy (XPS) measurements (METHODS), to an excellent extent, and this ordering is driven by the large difference in the electronic charges and ionic radii between $Sr^{2+}$ and $Os^{6+}$[21]. Consequently, the experimentally derived crystal structure does not allow for Os–O–Os paths. Therefore, advanced mechanisms need to be considered, since the Goodenough–Kanamori rules[22], which well predict magnetic interactions between two next-nearest-neighbor magnetic cations through a nonmagnetic anion, do not cover the theoretical framework for the FM order in $Sr_3OsO_6$.

**Magnetic properties.** Figure 2a, b shows the temperature dependence of magnetization versus the magnetic field ($M–H$) of a $Sr_3OsO_6$ film. The hysteretic response of the $Sr_3OsO_6$ film shows a soft magnetic behavior with the small coercive field of ~100 Oe at 1.9 K (Fig. 2b), and the saturation magnetization at 70,000 Oe (Fig. 2a) decreases with increasing temperature. The saturation magnetization persists up to 1000 K [limit of measurement range (Methods)], indicating $T_C > 1000$ K. Figure 2c shows the magnetization versus temperature ($M–T$) curve with $H = 2000$ Oe. In Fig. 2c, we also plot the spontaneous magnetization as a function of temperature. The $T_C$ value, estimated from the extrapolation of the $M−T$ curve to the zero magnetization axis, is about 1060 K (Fig. 2c). This is the highest $T_C$ among all insulators and oxides, and the highest magnetic ordering temperature in any compound without $3d$ transition elements[23].

Noteworthy, the saturation magnetization of $Sr_3OsO_6$ (~49 emu/cc at 1.9 K) is significantly smaller than that for typical magnetic metals; e.g., $Nd_2Fe_{14}B$ (~1280 emu/cc), $SmCo_5$ (~860 emu/cc) and Alnico 5 (~1120 emu/cc)[24], and typical ferrites; e.g., $CoFe_2O_4$ (~430 emu/cc), $Y_3Fe_5O_{12}$ (~170 emu/cc) and $LiFe_5O_8$ (~390 emu/cc)[6]. The small saturation magnetization, unique to $Sr_3OsO_6$, may encourage the development of spintronic devices utilizing small stray fields and low-energy spin-transfer-torque switching[25], which are advantageous for high-density integration and low-power consumption. This small saturation magnetization is, most likely, associated with the low composition ratio of Os in $Sr_3OsO_6$. The saturation magnetic moment of Os at 1.9 K was estimated to be 0.77 $\mu_B$/Os, which is smaller than the

expected value of the spin-only magnetic moment of 2 $\mu_B$ for the $Os^{6+}$ ($5d^2 t_{2g}^2$) state with $S = 1$. This apparent deviation requires that SOC has to be taken into account, which is often the case with $5d$ systems[18,26,27].

**Electrical properties.** While such high $T_C$ is common for systems with free charge carriers, e.g., $Fe_3O_4$ and Co, their absence in $Sr_3OsO_6$ requires other exchange mechanisms. The temperature dependence of resistivity ($\rho$) for a $Sr_3OsO_6$ film is shown in Fig. 3a. It increases with decreasing temperature and it exceeds the measurable range below 120 K. The electronic charge carriers [$5d$ electrons in the $Os^{6+}$ state] move by hopping between localized electronic states, and this is supported by $\ln(\rho) \propto T^{-1/4}$ [variable range hopping (VRH) model] (Fig. 3b) along with the high resistivity value [$\rho(300$ K$) = 75\ \Omega$ cm]. Other mechanisms, e.g., $\ln(\rho) \propto T^{-1/2}$ [Efros–Shklovskii Hopping (ESH) model] and $\ln(\rho) \propto T^{-1}$ [thermal activation (TA) model], are not supported by the electronic transport measurements. Figure 3c shows electron energy loss spectroscopy (EELS) spectra of a $Sr_3OsO_6$ film measured at three different positions as indicated in the cross-sectional STEM image (inset of Fig. 3c) with a spot size of ~4 nm. The EELS spectrum of a material corresponds to the loss function $Im(-1/\epsilon)$, where $\epsilon$ is a complex dielectric function. The three EELS spectra (Fig. 3c) taken of $Sr_3OsO_6$ are almost identical, indicating that electronic states are uniform in the entire $Sr_3OsO_6$ film. The optical bandgap (indicated by the black arrow), at which EELS intensities start to increase[28,29], is ~2.65 eV. Accordingly, models based on the double exchange interaction where itinerant electrons are driving the magnetic order, can be ruled out as the origin of ferromagnetic order in $Sr_3OsO_6$. In addition, the long distance between nearest Os atoms (5.81 Å) renders the possibility of direct exchange interactions unlikely[30,31].

**Electronic-structure calculations.** We analyzed the electronic and magnetic states of $Sr_3OsO_6$ by DFT with SOC (METHODS) using calculations with the Perdew–Burke–Ernzerhof (PBE) functional[32]. Canted FM order (Fig. 4a) was found to have the lowest total energy among many possible magnetic arrangements. However, the energy differences between the canted FM order and the collinear FM (Supplementary Fig. 10a), (001) AFM (Supplementary Fig. 10b), and (111) AFM orders (Supplementary Fig. 10c) are very small (~3.6 meV per atom, ~1.4 meV per atom and ~0.29 meV per atom, respectively), implying a competition among these orders. Note that canted magnetic orders have been reported in other Os containing double-perovskites[11,33]. The optical bandgap determined by GGA-PBE $+ U +$ SOC calculations of $Sr_3OsO_6$ with the canted FM order is ~0.69 eV, and this

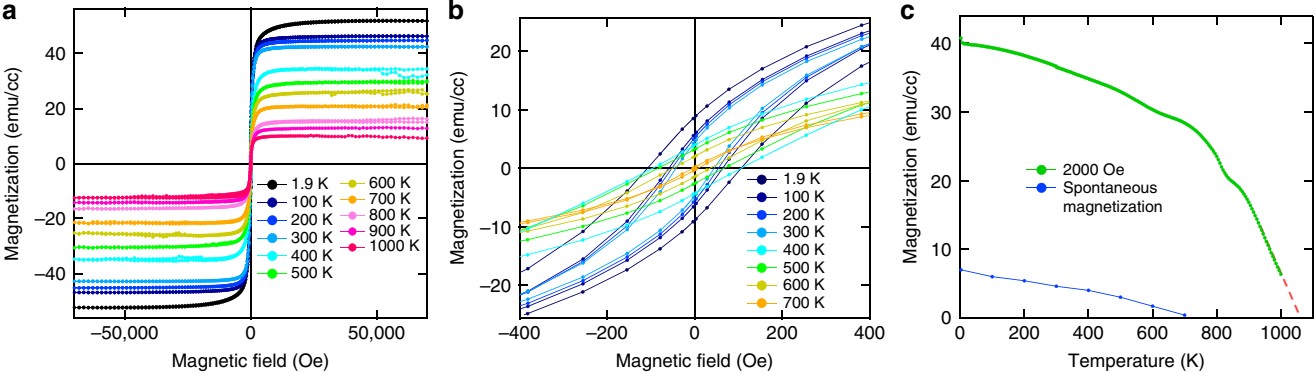

**Fig. 2** Magnetic properties of a $Sr_3OsO_6$ film. **a** In-plane $M–H$ curves at 1.9 up to 1000 K for a $Sr_3OsO_6$ film grown on (001) $SrTiO_3$. Here, $H$ was applied to the [100] direction. **b** Close-up near the zero magnetic field in **a**. **c** $M–T$ curve with $H = 2000$ Oe applied to the [100] direction for a $Sr_3OsO_6$ film grown on (001) $SrTiO_3$. Spontaneous magnetization deduced from Fig. 2b as a function of temperature is also shown

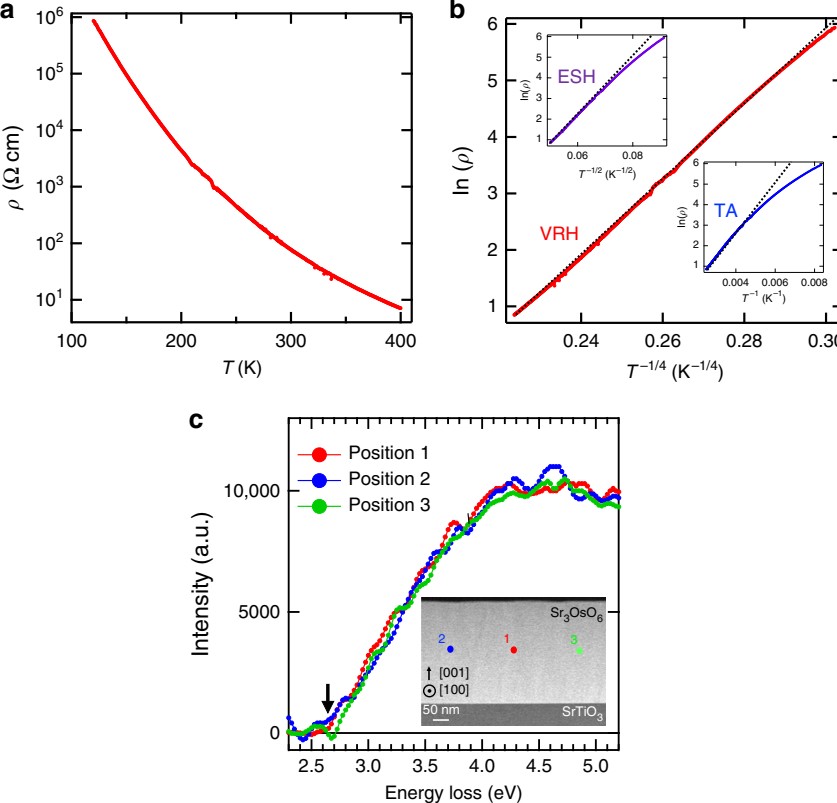

**Fig. 3** Resistivity and dielectric properties of a $Sr_3OsO_6$ film. **a** $\rho$–$T$ curve for a $Sr_3OsO_6$ film. **b** Logarithm of $\rho$ versus $T^{-1/4}$ plot, corresponding to the VRH model. The insets of **b** show the logarithm of $\rho$ versus $T^{-1/2}$ and $T^{-1}$ plots, corresponding to the ESH and TA models, respectively. The black dashed lines in **b** are guides to the eye. **c** The EELS spectra of a $Sr_3OsO_6$ film measured at the spots indicated in the cross-sectional STEM image (inset). The background was corrected with a power-law fit from 2 to 2.3 eV

corresponds to an insulating state (Fig. 4b, c). With $U$ + SOC, the $t_{2g\uparrow}$ states are split into effective total angular momenta $J_{eff} = 3/2$ (doublet) and $J_{eff} = 1/2$ (singlet) states while opening a gap. The $J_{eff} = 3/2$ states are fully occupied with two $5d$ electrons per $Os^{6+}$, resulting in the insulating state. Note that these calculations reveal a metallic ground state (i.e., no gap at the Fermi energy ($E_F$)), if $U = 0$ eV. This is because the band dispersions of $J_{eff} = 3/2$ and $J_{eff} = 1/2$ states are greater than the spin–orbit splitting. The calculated magnetic moment of osmium is $1.56\,\mu_B$/Os, which is closer to the experimentally determined value ($0.77\,\mu_B$/Os) as compared with the spin-only magnetic moment ($2\,\mu_B$/Os). We also calculated the element-specific partial density-of-state (PDOS) of $Sr_3OsO_6$ for the canted FM order by the Heyd–Scuseria–Ernzerhof (HSE) functional + SOC method (Supplementary Fig. 11) to properly implement the interaction length scale found in solids. Consequently, the DFT calculations with the HSE functional have superior prediction capabilities regarding band gaps of insulators[34]. The bandgap estimated by the HSE + SOC method (~1.41 eV) is closer to the experimentally obtained one (~2.65 eV) than that estimated by the GGA-PBE + $U$ + SOC calculations (~0.69 eV). Although further work is required to reveal the underlying mechanisms driving the FM order in $Sr_3OsO_6$, our calculations provide information on the magnetic arrangement at the ground state and how the energy gap is opened by the interplay between Coulomb repulsion ($U$) and SOC.

## Discussion

The origin of the robust ferromagnetism in this magnetic insulator ($Sr_3OsO_6$) remains murky, and it appears to be a challenge to reveal the underlying electronic exchange mechanisms. The extended superexchange paths (Os–O–Sr–O–Os), which are well recognized to drive the magnetic order in Os containing double-perovskites[30,31,35], are one possible origin of the ferromagnetism. Since a high magnetic ordering temperature through the extended superexchange paths requires a strong $p$–$d$ hybridization[35], further investigations into the electronic structures of $Sr_3OsO_6$, such as X-ray magnetic circular dichroism (XMCD), are required. Nevertheless, our DFT calculations with both the PBE (Fig. 4c) and HSE (Supplementary Fig. 11) functionals imply a non-negligible $p$–$d$ hybridization.

Attention should be paid to the excellent Os order, since cationic disorder deteriorates magnetic ordering in magnetic insulators[36–38]. Besides, $Ca_3OsO_6$ shows an antiferromagnetic order below 50 K[21], despite its high Os order and being isoelectronic to $Sr_3OsO_6$. A remarkable difference between $Sr_3OsO_6$ and $Ca_3OsO_6$ is their crystal structure ($Ca_3OsO_6$ is monoclinic with tilted $OsO_6$ octahedrons). Such a difference in magnetic order despite the isoelectronic structures was also reported for the isoelectronic pair $SrRuO_3$ (pseudo-cubic perovskite, ferromagnetic metal) and $CaRuO_3$ (orthorhombic perovskite, paramagnetic metal)[39], and the difference in the magnetic order in $SrRuO_3$ and $CaRuO_3$ is thought to be associated with the strength of the perovskite distortion. The network morphology hosting the mechanisms of exchange interactions is subject to such distortions, thus, likely driving $Ca_3OsO_6$ towards an antiferromagnetic instability. In addition, the cubic double-perovskite structure, in which only one $5d$ transition metal element occupies the B site, possesses 12 nearest neighbor magnetic ions in contrast to other $4d$ or $5d$ element-containing double-perovskites, in which both B and B′ sites are occupied by two different transition metal

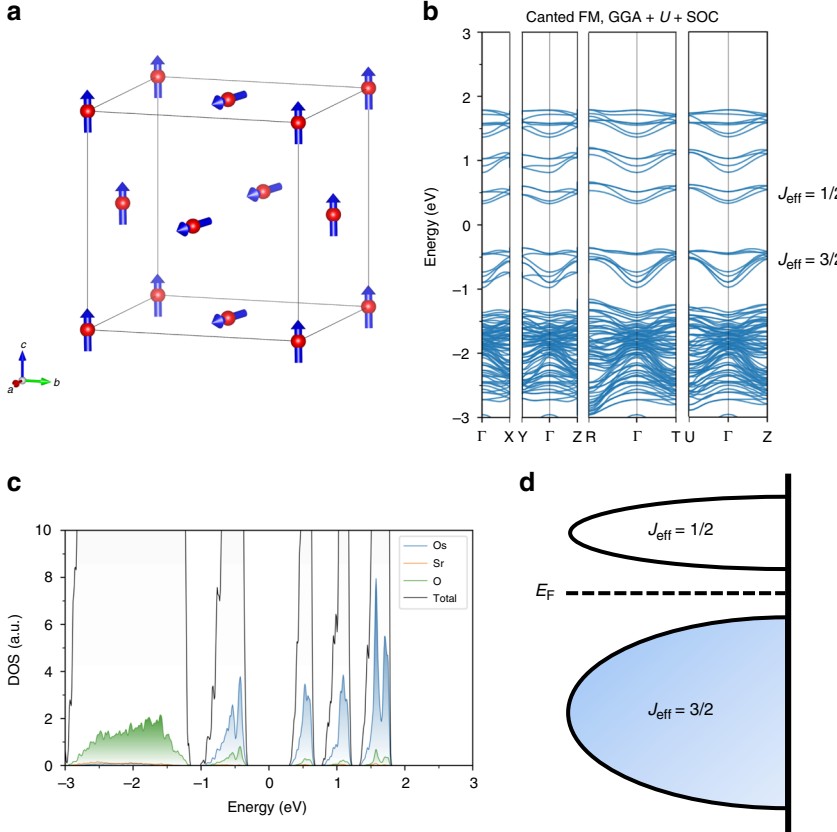

**Fig. 4** Electronic-structure calculations with the PBE functional. **a** Schematic diagram of the magnetic ground state (canted FM order) of $Sr_3OsO_6$ obtained from the DFT calculation. In **a**, red spheres and blue arrows indicate Os atoms and magnetic moments of Os atoms, respectively, and the Sr and O atoms are omitted for simplicity. **b** The band structures for $Sr_3OsO_6$ with the canted FM order calculated by GGA + $U$ + SOC. **c** The element-specific partial density-of-state (PDOS) for the canted FM order calculated by GGA + $U$ + SOC. In **c**, blue, orange, and green curves indicate the density-of-states per one Os, Sr, and O atoms, respectively. In **b** and **c**, the origin of energy was taken at the center of the bandgap. **d** Schematic energy diagrams for the Os $5d^2$ configurations. In **d**, only PDOS for Os is taken into account and the contributions by Sr and O are omitted for simplicity

elements, e.g., $Sr_2FeMoO_6$ ($T_C$ = 415 K)[7], $Sr_2CrReO_6$ ($T_C$ = 634 K)[9] and $Sr_2CrOsO_6$ ($T_C$ = 725 K)[13]. This could be one of the reasons for the robust ferromagnetism in $Sr_3OsO_6$ as predicted for halide double-perovskites[40]. Therefore, we suggest that the high Os order with the cubic structure plays an essential role in the robust ferromagnetism achieved in $Sr_3OsO_6$.

Our current findings in epitaxial $Sr_3OsO_6$ films—an extraordinarily high $T_C$ of 1060 K, $J_{eff}$ = 3/2 insulating state, rock-salt type $Os^{6+}$ order and small magnetic moment—enrich the family of ferri/ferromagnetic insulators. Although the underlying electronic exchange mechanisms driving the robust FM order in $Sr_3OsO_6$ remain murky, applications of $Sr_3OsO_6$ to oxide-electronics[41,42] beyond the current ferrite technology are feasible.

## Methods

**Growth of $Sr_3OsO_6$ on $SrTiO_3$.** We grew the high-quality epitaxial B-site ordered double-perovskite (001) $Sr_3OsO_6$ films (300- or 250-nm thick) on (001) $SrTiO_3$ substrates (CrysTec GmbH) in a custom-designed molecular beam epitaxy (MBE) system[43,44] (Supplementary Fig. 1a). After cleaning with $CHCl_3$ (10 min, two times) and acetone (5 min) by an ultrasonic cleaner, the $SrTiO_3$ substrate was introduced in the MBE growth chamber. After degassing the substrate at 400 °C for 30 min and successive thermal cleaning at 650 °C for 30 min, we grew a $Sr_3OsO_6$ film. The growth temperature was 650 °C. The oxidation during the growth was carried out with $O_3$ gas (non-distilled, ~10% concentration) from a commercial ozone generator as a flow rate of 2 sccm. After the growth, films were cooled to room temperature under ultra-high vacuum (UHV). The MBE system is equipped with multiple e-beam evaporators (Hydra, Thermionics) for Sr and Os. Note that to avoid a formation of $OsO_4$, which is highly toxic, we used Os chunks with a radius of about 3 mm (not Os powder), which are stable in air at room temperature. The electron impact emission spectroscopy (EIES) sensor (Guardian, Inficon) is located next to the sample heater in the MBE in the same horizontal plane. The

sensor head is equipped with a filament, which generates thermal electrons for the excitation of Sr and Os atoms. Optical band-pass filters are used for element-specific detection of the excited optical signals, since the emitted light spectra are characteristic for Sr and Os. The EIES sensor is equipped with photomultipliers (PMTs) located outside of the vacuum chamber that convert optical signals into electrical signals. The Sr and Os fluxes measured by EIES were kept constant (Supplementary Fig. 1b) by the proportional-integral-derivative (PID) control of the evaporation source power supply. We optimized the flux ratio of Sr and Os to ascertain $Sr_3OsO_6$ films with a high saturation magnetic moment. Supplementary Fig. 2 shows the in-plane $M–H$ curves at 300 K for $Sr_3OsO_6$ films grown with different flux ratios of Sr and Os. The saturation magnetic moment of the film grown with the flux ratio of Sr:Os = 2.05:1 is ten or more times larger than those for the films grown with the flux ratio of Sr:Os = 2.05:1.02 and 2.05:0.98. This means that the magnetic properties of $Sr_3OsO_6$ films are very sensitive to the Sr/Os ratio and that well-controlled Sr and Os fluxes during the growth are important for the high saturation magnetic moment. Therefore, in this study, we set the flux ratio of Sr:Os at 2.05:1.

The cubic crystal structure of $Sr_3OsO_6$ is illustrated in Supplementary Fig. 3a. Supplementary Fig. 3b,c show reflection high-energy electron diffraction (RHEED) patterns of a $Sr_3OsO_6$ thin film surface, where the sharp streaky patterns with clear surface reconstruction indicate the growth of the $Sr_3OsO_6$ film in a layer-by-layer manner, leading to the high crystalline quality of the film. Notably, [01$l$] diffractions are not seen (Supplementary Fig. 3b) due to extinction rules, indicating the formation of a cubic B-site ordered double-perovskite[8,12,20]. The cubic structure model is further supported by high-resolution X-ray reciprocal space mapping (HRRSM) (Supplementary Fig. 3d): the in-plane and out-of-plane lattice constants of $Sr_3OsO_6$ are identical within the resolution limits (8.24 ± 0.03 and 8.22 ± 0.03 Å, respectively). It is therefore reasonable that the $Sr_3OsO_6$ films are epitaxially but not coherently grown on the $SrTiO_3$ (3.905 Å) substrate.

**Transmission electron microscopy and transmission electron diffraction.** High-angle annular dark-field (HAADF), annular bright-field (ABF) scanning transmission electron microscopy (STEM) images, and transmission electron diffraction (TED) patterns were taken with a JEOL JEM-ARM 200F microscope.

Electron energy loss spectroscopy (EELS) spectra of a $Sr_3OsO_6$ film were recorded from three spots with a ~4-nm diameter also with a JEOL JEM-ARM 200F microscope.

Supplementary Figs. 3e–g show cross-sectional HAADF-STEM images of a $Sr_3OsO_6$ film taken along the [100] direction. At a glance, one can recognize that a single-crystalline $Sr_3OsO_6$ film with an abrupt substrate/film interface has been grown epitaxially on a (001) $SrTiO_3$ substrate, as expected from the RHEED. Misfit dislocations at the $Sr_3OsO_6/SrTiO_3$ interface (Supplementary Fig. 3g) are due to the ~5% larger lattice constant of the perovskite $Sr_3OsO_6$ lattice (8.23 Å/2 = 4.115 Å) than that of $SrTiO_3$ (3.905 Å). The cubic crystal structure of $Sr_3OsO_6$ was also confirmed by the STEM analysis. In addition to the [100] direction (Supplementary Figs. 3e–g), the epitaxial growth of the $Sr_3OsO_6$ layer on the $SrTiO_3$ substrate was also confirmed by STEM images taken along the [110] direction (Supplementary Figs 4a–c). The rock-salt type order of $Os^{6+}$ (Fig. 1) is observed to an excellent extent (Supplementary Figs 4d,e).

Supplementary Fig. 5 shows the TED pattern for a $Sr_3OsO_6$ film taken along the [110] direction. The diffraction pattern agrees very well with the calculated diffraction pattern for the ideal cubic B-site ordered double-perovskite structure shown in Supplementary Fig. 3a. The extinction rules for a fully B-site ordered double-perovskite demand that either even (hkl) or odd (hkl) peaks are permitted[20] —and that is exactly what is seen here, confirming the cubic B-site ordered double-perovskite structure.

**Chemical composition of a $Sr_3OsO_6$ film.** Supplementary Fig. 6a shows the depth profile of the chemical composition of a $Sr_3OsO_6$ film (250-nm thick) estimated from Rutherford backscattering spectroscopy (RBS). The chemical composition of the $Sr_3OsO_6$ layer is uniform (Sr:Os:O = 2.7 ± 0.1:1.15 ± 0.05:6.15 ± 0.4). The concentrations of Os and Oxygen are slightly larger than those for an ideal composition (Sr:Os:O = 3:1:6). This difference may originate from the non-stoichiometry and existence of a small amount of paramagnetic metallic $OsO_2$[45–47], which was observed in the X-ray diffraction (XRD) measurements, as described below. To exclude the possibility of the contamination by magnetic impurities, we performed EDS measurement for a $Sr_3OsO_6$ film (Supplementary Fig. 6b). There are no other peaks except for Sr, Os, Ti, O, and C, which confirms the absence of magnetic impurities.

**X-ray diffraction.** We performed $\theta$–$2\theta$ and reciprocal space map XRD measurements of the $Sr_3OsO_6$ films with a Bruker D8 diffractometer using monochromatic Cu $K\alpha_1$ radiation at room temperature. In Supplementary Fig. 6c, we show the $\theta$–$2\theta$ XRD pattern for a $Sr_3OsO_6$ film on (001) $SrTiO_3$. In addition to the diffraction peaks of the $SrTiO_3$ substrates, (002) and (004) diffractions from $Sr_3OsO_6$ are clearly observed. No (001) and (003) diffractions from $Sr_3OsO_6$ are seen due to the extinction rules. Note that traces of $OsO_2$, which is known as a paramagnetic metal[45–47], are detected as indicated by *. The XRD intensities of $OsO_2$ are about 700 times smaller than those of $Sr_3OsO_6$, and segregation of $OsO_2$ is not discernible in the STEM images, indicating that the volume fraction of $OsO_2$ (paramagnetic metal[45–47]) is negligible. Therefore, $Sr_3OsO_6$ dominates the magnetic response of the film.

**X-ray photoemission measurements.** X-ray photoemission spectroscopy (XPS) is one of the most powerful methods to determine the valence of Os in compounds[48,49], since the $4f_{7/2}$ core level binding energies in Os compounds with well-defined oxidation states are known. An ULVAC-PHI Model XPS5700 with a monochromatized Al $K\alpha$ (1486.6 eV) source operated at 200 W was used for the experiments. The scale of binding energy was calibrated against the C 1s line (284.6 eV). Supplementary Fig. 7 shows the Os 4f spectrum of a $Sr_3OsO_6$ film at 300 K. The observed $4f_{7/2}$ binding energy (54.1 eV) is close to the reported values for those of $Os^{6+}$ states (53.2–53.8 eV) and far from those for $Os^{2+}$ (49.7 eV), $Os^{3+}$ states (50.4–51.0 eV), $Os^{4+}$ states (51.7–52.3 eV), and $Os^{8+}$ states (55.9–56.3 eV)[48,49]. Accordingly, the hexavalent state of Os ($Os^{6+}$) is supported. Note that a shoulder structure at ~53 eV may originate from a surface layer formed due to the slightly hygroscopic nature of $Sr_3OsO_6$ because the sample was transferred to the XPS apparatus in atmosphere.

**Resistivity measurements.** Resistivity was measured using the four-probe method in a Physical Property Measurement System (PPMS) Dynacool sample chamber. The Ag electrodes were deposited on a $Sr_3OsO_6$ surface and connected to an Agilent 3458A Multimeter.

**Magnetic measurements.** The magnetization measurements for $Sr_3OsO_6$ films were performed with a Quantum Design MPMS3 SQUID-VSM magnetometer. Using a quartz sample holder (oven sample holder), we measured the $M$–$T$ curves while increasing the temperature from 1.9 (300) to 300 (1000) K with $H = 2000$ Oe applied along the [100] or [110] direction. In the $M$–$T$ measurements, $M$ was measured with increasing temperature after the sample was cooled to 1.9 (300) K from 300 (1000) K without a magnetic field. We also measured $M$–$H$ curves at 1.9–300 K (400–1000 K) using the quartz sample holder (oven sample holder).

To check the accuracy of the measurement temperature in the MPMS SQUID-VSM magnetometer, we measured the magnetic properties of a pure Ni reference

plate (Quantum Design Part Number: 4505–155). The $M$–$H$ curves at 300 and 1000 K show FM and paramagnetic response, respectively (Supplementary Fig. 8a), and the magnetization of Ni rapidly increases between 623 and 629 K (Supplementary Fig. 8b). These results indicate that the $T_C$ of Ni is between 623 and 629 K. This is consistent with the $T_C$ value in the literature (627 K)[50]. Thus, the error in the measurement temperature in the MPMS SQUID-VSM magnetometer is less than ±4 K.

Supplementary Fig. 8c shows the in-plane $M$–$H$ curves at 1.9, 300, and 1000 K of a $SrTiO_3$ substrate. They show only a linear diamagnetic response at 300 and 1000 K. The nonlinear magnetic response near the zero magnetic field at 1.9 K indicates the existence of paramagnetic impurities in the $SrTiO_3$ substrates. In Fig. 2, the linear diamagnetic response of the magnetic moment for the $SrTiO_3$ substrate was subtracted from the raw $M(H)$ and $M(T)$.

Supplementary Fig. 8d shows the $M$–$T$ curve with $H = 2000$ Oe for the oven sample holder without a sample. The curve shows a dip structure at around 800 K. This means that the dip structure in the $M$–$T$ curve at around 800 K for the $Sr_3OsO_6$ film (Fig. 2c) is an unavoidable experimental artifact.

The magnetic properties of $Sr_3OsO_6$ at 300 K did not change much after it was heated to 1000 K as shown in Supplementary Fig. 9a. This means that heating to 1000 K does not affect much its magnetic properties.

Although the $Sr_3OsO_6$ films were epitaxially grown on the $SrTiO_3$ substrates, the shapes of the in-plane $M$−$H$ curves measured with $H$ applied to the [100] or [110] direction are identical (Supplementary Fig. 9b). This indicates that the in-plane magnetic anisotropy of the $Sr_3OsO_6$ film is negligibly small. This small magnetic anisotropy might be related to the misfit dislocations (Supplementary Fig 3e–g), which often decrease the magnetic anisotropy of magnetic insulators[36–38].

**The electronic-structure calculations.** The electronic-structure calculations were based on density functional theory (DFT). The calculations were performed by using the Vienna Ab initio Simulation Package (VASP)[51,52] with the projector augmented-wave (PAW)[53,54] method and the Perdew–Burke–Ernzerhof (PBE) functional[32] within the generalized gradient approximation (GGA)[55]. To describe the localization of Os 5d electrons accurately, we used the DFT + U calculations[56]. The value of the screened Coulomb interaction $U = 3$ eV was used for the Os atoms. This value is comparable to reported values for Os containing double-perovskites (2–4 eV)[14,18,19,31]. The contribution of the spin–orbit coupling (SOC) was also included in our calculations. The crystal structure was optimized for the conventional unit cell (40 atoms) of $Sr_3OsO_6$ whose lattice constant was fixed to the experimental value 8.23 Å. We performed the optimization until all forces on the atoms become smaller than $10^{-5}$ eV/Å with a Γ-centered $2 \times 2 \times 2$ k-point grid and cut-off energy of 800 eV. The total energies and electronic structures were calculated with the optimized crystal structures resulting in the canted FM ground state (Fig. 4a).

By comparing the total energy of the magnetic ground state (canted FM order) (Fig. 4a) with those of the collinear FM order (Supplementary Fig. 10a), the (001) AFM order (Supplementary Fig. 10b) and the (111) AFM order (Supplementary Fig. 10c), we found that the energy differences between the canted FM order and the other orders are very small (~3.6, ~1.4, and ~0.29 meV per atom, respectively), implying a competition among these orders.

We also calculated the element-specific partial density-of-state (PDOS) of $Sr_3OsO_6$ for the canted FM order by the Heyd–Scuseria–Ernzerhof (HSE) + SOC method (Supplementary Fig. 11).

## Data availability

The data that support the plots in this paper and other findings of this study are available from the corresponding author upon request.

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

## Acknowledgements

We thank Ken-ichi Sasaki for a valuable discussion, Ai Ikeda for her help with the X-ray diffraction and resistivity measurements, Hiroshi Irie for his help with the resistivity measurements, and Kazuhide Kumakura for his help with the MPMS3 SQUID-VSM oven option.

## Author contributions

Y.K. and Y.K.W. prepared the samples. Y.K.W. performed experimental measurements and data analysis. N.T. and S.T. carried out the electronic-structure calculations. Y.K.W. wrote the paper. Y.K.W., Y.K., N.T., T.B., S.T., Y.T., and H.Y. contributed to the manuscript and the interpretation of the data.

## Additional information

**Competing interests:** The authors declare no competing interests.

