## [Peer Review File · Nature Communications]

Reviewers' Comments:

Reviewer #1:

Remarks to the Author:

This is another example of the extraordinary properties of 4d and 5d transition metal oxides (see work on SrTcO₃) and it is likely to attract significant interest from both theoreticians and experimentalists. The experimental work appears to be well conducted and the results are well described and novel.

I have a couple of significant concerns with this work - no where do the authors actually demonstrate that Sr₃OsO₆ has an ordered cubic structure. Based on the work on the prior work on Ca₃OsO₆ I have no doubt that the Sr and Os cations order on the perovskite B-site but I suspect the symmetry will be lower. This is a critical point since the overlap of the Os 5d orbitals (Os-O-Sr-O-Os) and hence electronic properties will be dependent on the angles.

Secondly the authors should address the significant difference in the electronic properties of their thin films of Sr₃OsO₆ and of bulk Ca₃OsO₆. Starting such a discussion with the difference between the isostructural pair SrRuO₃ (ferromagnetic) and CaRuO₃ (non-magnetic) would be logical.

I suspect to understand the origin of the observed high Neel temperature it will be necessary to study a bulk sample and that the preparation of this may well require specialist high pressure methods. The current paper will stimulate such studies.

In summary I support publication of the work as a communication in the belief that this will serve as a catalyst to future work provided the authors can address my concern about symmetry

Reviewer #2:

Remarks to the Author:

Ferromagnetic insulators that can work at room temperature are essential for spintronics. In this paper, the authors investigated the properties of Sr₃OsO₆ by experiments and DFT simulations. They find Sr₃OsO₆ is an insulator, and the Curie temperature of Sr₃OsO₆ can reach as high as 1,000 K, which is the highest TC among all insulators. In total, this is a well-structured and logical paper containing interesting results which deserves publication in Nature Communications. However, before that, the authors should deal with a few points raised below:

1. The mechanism of this work is partial agree with a recent DFT investigation about halide double perovskite (doi: 10.1039/c8mh00590g). This reference is missing in this article. Compared to Sr₂FeMoO₆, Sr₂CrReO₆ and Sr₂CrOsO₆, the alternative arrangement of magnetic and nonmagnetic octahedrons can enlarge the nearest neighboring number of magnetic ions. Hence, spin flip needs to cross a higher barrier if exchange integral J is not too small, thereby leading a higher TC.
2. The AFM order is incomplete in DFT simulation. Authors only consider the AFM order of (1 0 0) surface. Please compared the energy of canted FM order and other AFM order, such as (1 1 1).
3. PBE functional usually underestimates the bandgaps. The bandgap of experiment (2.65 eV) and calculation (0.37 eV) also demonstrate this problem in this article. Please use HSE method to correct the bandgap and it will make your DFT results more reliable.
4. The superexchange path Os-O-O-Os is not suitable for this cubic double-perovskite, namely 2116 structure (316 structure in this work). Please delete this path in page 5. In fact, the crystal structure of Ca₃LiOsO₆ in reference 23 is not double perovskite. It is a derivative of hexagonal phase perovskite, namely 416 structure. The octahedrons in 416 structure are isolated, hence Os-O-O-Os path is the only choice for the exchange coupling. However, the O atoms in 316 structure prefer to connect with Sr and Os cations which display positive valency. Hence, the direct interaction between O-O is so weak that we can ignore it.

Reviewer #3:
None

Reviewers' comments

Reviewer #1 (Remarks to the Author):

This is another example of the extraordinary properties of 4d and 5d transition metal oxides (see work on SrTcO₃) and it is likely to attract significant interest from both theoreticians and experimentalists. The experimental work appears to be well conducted and the results are well described and novel.

I have a couple of significant concerns with this work - no where do the authors actually demonstrate that Sr₃OsO₆ has an ordered cubic structure. Based on the work on the prior work on Ca₃OsO₆ I have no doubt that the Sr and Os cations order on the perovskite B-site but I suspect the symmetry will be lower. This is a critical point since the overlap of the Os 5d orbitals (Os-O-Sr-O-Os) and hence electronic properties will be dependent on the angles.

Secondly the authors should address the significant difference in the electronic properties of their thin films of Sr₃OsO₆ and of bulk Ca₃OsO₆. Starting such a discussion with the difference between the isostructural pair SrRuO₃ (ferromagnetic) and CaRuO₃ (non-magnetic) would be logical.

I suspect to understand the origin of the observed high Neel temperature it will be necessary to study a bulk sample and that the preparation of this may well require specialist high pressure methods. The current paper will stimulate such studies.

In summary I support publication of the work as a communication in the belief that this will serve as a catalyst to future work provided the authors can address my concern about symmetry

Reviewer #2 (Remarks to the Author):

Ferromagnetic insulators that can work at room temperature are essential for spintronics. In this paper, the authors investigated the properties of Sr₃OsO₆ by experiments and DFT simulations. They find Sr₃OsO₆ is an insulator, and the Curie temperature of Sr₃OsO₆ can reach as high as 1,000 K, which is the highest TC among all insulators. In total, this is a well-structured and logical paper containing interesting results which deserves publication in Nature Communications. However, before that, the authors should deal with a few points raised below:

1. The mechanism of this work is partial agree with a recent DFT investigation about

halide double perovskite (doi: 10.1039/c8mh00590g). This reference is missing in this article. Compared to $\text{Sr}_2\text{FeMoO}_6$, $\text{Sr}_2\text{CrReO}_6$ and $\text{Sr}_2\text{CrOsO}_6$, the alternative arrangement of magnetic and nonmagnetic octahedrons can enlarge the nearest neighboring number of magnetic ions. Hence, spin flip needs to cross a higher barrier if exchange integral J is not too small, thereby leading a higher TC.

2. The AFM order is incomplete in DFT simulation. Authors only consider the AFM order of (1 0 0) surface. Please compared the energy of canted FM order and other AFM order, such as (1 1 1).

3. PBE functional usually underestimates the bandgaps. The bandgap of experiment (2.65 eV) and calculation (0.37 eV) also demonstrate this problem in this article. Please use HSE method to correct the bandgap and it will make your DFT results more reliable.

4. The superexchange path Os-O-O-Os is not suitable for this cubic double-perovskite, namely 2116 structure (316 structure in this work). Please delete this path in page 5. In fact, the crystal structure of $\text{Ca}_3\text{LiOsO}_6$ in reference 23 is not double perovskite. It is a derivative of hexagonal phase perovskite, namely 416 structure. The octahedrons in 416 structure are isolated, hence Os-O-O-Os path is the only choice for the exchange coupling. However, the O atoms in 316 structure prefer to connect with Sr and Os cations which display positive valency. Hence, the direct interaction between O-O is so weak that we can ignore it.

Response to Reviewers' comments

The point-by-point responses to the reviewers' comments are given below.

Response to the comments of Reviewer #1

Comment by Reviewer #1 (1):

I have a couple of significant concerns with this work - no where do the authors actually demonstrate that Sr₃OsO₆ has an ordered cubic structure. Based on the work on the prior work on Ca₃OsO₆ I have no doubt that the Sr and Os cations order on the perovskite B-site but I suspect the symmetry will be lower. This is a critical point since the overlap of the Os 5d orbitals (Os-O-Sr-O-Os) and hence electronic properties will be dependent on the angles.

Response:

As described in the previous/revised manuscript and METHODS, high-resolution scanning transmission electron microscopy (STEM), combined with high-resolution reciprocal space mapping (HRRSM) and reflection high-energy electron diffraction (RHEED), ascertained a cubic B-site ordered double-perovskite structure. In particular, the B-site ordering was clearly observed by atomic-resolution STEM to an excellent extent (see Fig. 1 and Supplementary Fig. 4 in the revised manuscript). To confirm the cubic B-site ordered double-perovskite structure further, we carried out transmission-electron diffraction (TED) measurement.

Figure R1 shows TED pattern for a Sr₃OsO₆ film taken along the [110] direction. The diffraction pattern agrees very well with the calculated diffraction pattern for the ideal cubic B-site ordered double-perovskite structure shown in Supplementary Fig. 3a in the revised manuscript. The extinction rules for a fully B-site ordered double perovskite demand that either even (*hkl*) or odd (*hkl*) peaks are permitted^{R1} – and that is exactly what is seen here, confirming the cubic B-site ordered double-perovskite structure.

We described these points in the revised METHODS (page 18, line 4).

Figure R1. TED pattern for a Sr_3OsO_6 film taken along the $[110]$ axis. The red dashed circles represent calculated diffraction pattern obtained by the Fourier transform of the ideal cubic B-site ordered double-perovskite structure.

R1. Manako, T., Izumi, M., Konishi, Y., Kobayashi, K. I., Kawasaki, M. & Tokura, Y. Epitaxial thin films of ordered double perovskite $\text{Sr}_2\text{FeMoO}_6$. *Appl. Phys. Lett.* **74**, 2215 (1999).

Comment by Reviewer #1 (2):

Secondly the authors should address the significant difference in the electronic properties of their thin films of Sr_3OsO_6 and of bulk Ca_3OsO_6 . Starting such a discussion with the difference between the isostructural pair SrRuO_3 (ferromagnetic) and CaRuO_3 (non-magnetic) would be logical.

Response:

Although the origin of the robust ferromagnetism in Sr_3OsO_6 remains murky, attention should be paid to the excellent Os order, since cationic disorder deteriorates magnetic ordering in magnetic insulators^{R2,R3}. Besides, Ca_3OsO_6 shows an antiferromagnetic order only below 50 K^{R4}, despite its high Os order and while being isoelectronic to Sr_3OsO_6 . A remarkable difference between Sr_3OsO_6 and Ca_3OsO_6 is their crystal structures (Ca_3OsO_6 has a monoclinic structure, and the OsO_6 octahedrons are tilted). As the reviewer pointed out, such a difference in magnetic order despite the

isoelectronic structures was also reported for the isoelectronic pair SrRuO₃ (pseudo-cubic perovskite, ferromagnetic metal) and CaRuO₃ (orthorhombic perovskite, paramagnetic metal)^{R5}, and the difference in the magnetic order in SrRuO₃ and CaRuO₃ is thought to be associated with the strength of the perovskite distortion. The network morphology hosting the mechanisms of exchange interactions is subject to such distortions and thus likely drives Ca₃OsO₆ towards an antiferromagnetic instability. Therefore, we suggest that the high Os order with the cubic structure plays an essential role in the robust ferromagnetism achieved in Sr₃OsO₆.

We described these points in the revised manuscript (page 6, line 14).

- R2. Margulies, D. T. *et al.* Origin of the anomalous magnetic behavior in single crystal Fe₃O₄ films. *Phys. Rev. Lett.* **79**, 5162 (1997).
- R3. Wakabayashi, Y. K. *et al.* Electronic structure and magnetic properties of magnetically dead layers in epitaxial CoFe₂O₄/Al₂O₃/Si(111) films studied by X-ray magnetic circular dichroism. *Phys. Rev. B* **96**, 104410 (2017).
- R4. Feng, H. L. *et al.* High-pressure crystal growth and electromagnetic properties of 5d double-perovskite Ca₃OsO₆. *J. Solid State Chem.* **201**, 186 (2013).
- R5. Koster, G. *et al.* Structure, physical properties, and applications of SrRuO₃ thin films. *Rev. Mod. Phys.* **84**, 253 (2012).

Response to the comments of Reviewer #2

Comment by Reviewer #2 (1):

1. The mechanism of this work partially agrees with a recent DFT investigation about halide double perovskite (doi: 10.1039/c8mh00590g). This reference is missing in this article. Compared to Sr₂FeMoO₆, Sr₂CrReO₆ and Sr₂CrOsO₆, the alternative arrangement of magnetic and nonmagnetic octahedrons can enlarge the nearest neighboring number of magnetic ions. Hence, spin flip needs to cross a higher barrier if exchange integral J is not too small, thereby leading a higher TC.

Response:

Thank you for drawing our attention to this exiting article which is, of course, included in the revised manuscript. As the reviewer pointed out, the cubic double perovskite structure, in which only one 5d transition metal element occupies the B sites, possesses a larger number of nearest neighboring magnetic ions of 12 than those in other 4d or 5d element-containing double-perovskites, in which both B and B' sites are occupied by two different transition metal elements, e.g., Sr₂FeMoO₆ ($T_C = 415$ K),

$\text{Sr}_2\text{CrReO}_6$ ($T_C = 634$ K) and $\text{Sr}_2\text{CrOsO}_6$ ($T_C = 725$ K). This could be, indeed, a strong reason for the emergence of the robust ferromagnetism in Sr_3OsO_6 as predicted in the halide double perovskites^{R6}.

We described these points in the revised manuscript (page 6, line 23).

R6. Cai, B. *et al.* A class of Pb-free double perovskite halide semiconductors with intrinsic ferromagnetism, large spin splitting and high Curie temperature. *Mater. Horiz.* **5**, 961 (2018).

Comment by Reviewer #2 (2):

2. The AFM order is incomplete in DFT simulation. Authors only consider the AFM order of (1 0 0) surface. Please compared the energy of canted FM order and other AFM order, such as (1 1 1).

Response:

In the revised version of our manuscript, we have, of course, also compared the energy of the canted FM order and (111) AFM order. By comparing the total energy of the magnetic ground state (canted FM order) (Fig. 4a in the revised manuscript) with those of the collinear FM order (Fig. R2a), (001) AFM order (Fig. R2b) and (111) AFM order (Fig. R2c), we found that the energy differences between the canted FM order and the other orders are very small (~ 3.6 meV per atom, ~ 1.4 meV per atom and ~ 0.29 meV per atom, respectively), implying a competition among these orders.

We described these points in the revised manuscript (page 5, line 8) and METHODS (page 21, line 11).

Figure R2. Schematic diagram of the magnetic orders. **a**, **b**, **c**, Schematic diagram of the collinear FM order (**a**), (001) AFM order (**b**) and (111) AFM order (**c**). In the figures, red spheres and blue arrows indicate Os atoms and Os magnetic moments, respectively, and the Sr and O atoms are omitted for the simplicity.

Comment by Reviewer #2 (3):

3. PBE functional usually underestimates the bandgaps. The bandgap of experiment (2.65 eV) and calculation (0.37 eV) also demonstrate this problem in this article. Please use HSE method to correct the bandgap and it will make your DFT results more reliable.

Response:

As the reviewer pointed out, generally, the DFT calculations with the Heyd–Scuseria–Ernzerhof (HSE) functional has superior prediction capabilities regarding band gaps of insulators^{R7}. We recalculated element-specific partial density-of-state (PDOS) of Sr₃OsO₆ for the canted FM order by the GGA-PBE + *U* + SOC method with a Γ -centered $7 \times 7 \times 7$ *k*-point grid to increase the accuracy of our calculations (Fig. 4 in the revised manuscript). After that, we also calculated the element-specific PDOS of Sr₃OsO₆ for the canted FM order by the HSE + SOC method (Fig. R3) to predict the band gap more correctly. The band gap estimated by the HSE + SOC method (~ 1.41 eV) is closer to the experimentally obtained band gap (~ 2.65 eV) than that estimated by the GGA-PBE + *U* + SOC calculations (~ 0.69 eV).

We described these points in the revised manuscript (page 5, line 21) and METHODS (page 21, line 16).

Figure R3. The element-specific PDOS for the canted FM order calculated by the HSE + SOC method. Here, blue, orange and green curves indicate the density-of-states per one Os, Sr and O atoms, respectively.

R7. Tran, F. & Blaha, P. Accurate band gaps of semiconductors and insulators with a semilocal exchange-correlation potential. Phys. Rev. Lett. **102**, 226401 (2009).

Comment by Reviewer #2 (4):

4. The superexchange path Os-O-O-Os is not suitable for this cubic double-perovskite, namely 2116 structure (316 structure in this work). Please delete this path in page 5. In fact, the crystal structure of $\text{Ca}_3\text{LiOsO}_6$ in reference 23 is not double perovskite. It is a derivative of hexagonal phase perovskite, namely 416 structure. The octahedrons in 416 structure are isolated, hence Os-O-O-Os path is the only choice for the exchange coupling. However, the O atoms in 316 structure prefer to connect with Sr and Os cations which display positive valency. Hence, the direct interaction between O-O is so weak that we can ignore it.

Response:

As the reviewer pointed out, the crystal structures of $\text{Ca}_3\text{LiOsO}_6$, Sr_2BOsO_6 ($B = \text{Y}$, In , Sc) and $\text{Sr}_2\text{ScOsO}_6$, in Refs. 26, 27 and 35 in the revised manuscript, are not cubic double perovskites. The symmetries of these oxides are lower than that of Sr_3OsO_6 , and the direct interaction between O-O in Sr_3OsO_6 is weak. Therefore, we have deleted the superexchange path Os-O-O-Os in the revised manuscript (page 6, line 8).

Reviewers' Comments:

Reviewer #1:

Remarks to the Author:

The authors have done a good job in addressing the concerns I raised in my original review of this paper and I am satisfied that this work is now suitable for publication as a Nature Communication

Reviewer #2:

Remarks to the Author:

The authors have addressed my concerns raised in the previous review and revised the manuscript accordingly. These results stimulate experimental works on constructing novel spintronic devices using Sr₃OsO₆ in view of its large spin-orbit coupling and high Curie temperature. I recommend publication of this manuscript in Nature Communications.